# Ultrasensitive proteome analysis using paramagnetic bead technology

Christopher S Hughes, Sophia Foehr, David A Garfield, Eileen E Furlong, Lars M Steinmetz & Jeroen Krijgsveld*

## Abstract

In order to obtain a systems-level understanding of a complex biological system, detailed proteome information is essential. Despite great progress in proteomics technologies, thorough interrogation of the proteome from quantity-limited biological samples is hampered by inefficiencies during processing. To address these challenges, here we introduce a novel protocol using paramagnetic beads, termed Single-Pot Solid-Phase-enhanced Sample Preparation (SP3). SP3 provides a rapid and unbiased means of proteomic sample preparation in a single tube that facilitates ultrasensitive analysis by outperforming existing protocols in terms of efficiency, scalability, speed, throughput, and flexibility. To illustrate these benefits, characterization of 1,000 HeLa cells and single *Drosophila* embryos is used to establish that SP3 provides an enhanced platform for profiling proteomes derived from sub-microgram amounts of material. These data present a first view of developmental stage-specific proteome dynamics in *Drosophila* at a single-embryo resolution, permitting characterization of inter-individual expression variation. Together, the findings of this work position SP3 as a superior protocol that facilitates exciting new directions in multiple areas of proteomics ranging from developmental biology to clinical applications.

**Keywords** mass spectrometry; paramagnetic beads; proteomics; quantification; sample preparation

**Subject Categories** Methods & Resources; Post-translational Modifications, Proteolysis & Proteomics

**Mol Syst Biol.** (2014) 10: 757

See also: **E Kanshin & P Thibault** (October 2014)

## Introduction

Diversity and complexity in cellular proteomes have driven the development of a broad range of protocols to improve analyses by mass spectrometry (MS). In traditional bottom-up experiments, these methods are optimized to enhance the depth of proteome coverage through generation of conditions favorable for proteolytic digestion and sample recovery (León *et al*, 2013; Tanca *et al*, 2013) and have led to charting of the near-complete proteomes of various mammalian cell lines (Beck *et al*, 2011; Moghaddas Gholami *et al*, 2013; Branca *et al*, 2014). However, performance in proteomic experiments drops steeply when protein amounts are limited, due to inefficiencies related to sample processing and instrument sensitivity. Although recent innovations in mass spectrometric instrumentation have accelerated the speed and sensitivity of proteome analysis (Hebert *et al*, 2014), further improvements can be obtained by emphasizing the optimization, simplification, and miniaturization of sample preparation.

To enhance sample processing, agents that aid in cell disruption and solubilization such as detergents and chaotropes are often utilized. Problematically, the majority of these additives are incompatible with proteolysis and MS analysis and thus necessitate removal using ultrafiltration (Wisniewski *et al*, 2009) and bead-based (Bereman *et al*, 2011; Hengel *et al*, 2012) or precipitation approaches, each of which increase handling and subsequent loss of material. As MS-based proteomics drives toward the analysis of rare and quantity-limited samples, ultrasensitive workflows that eliminate these losses are essential (Altelaar & Heck, 2012). This has led to the development of methodologies that minimize handling and promote high sample recovery (Ethier & Hou, 2006; Umar *et al*, 2007; Waanders *et al*, 2009; Wang *et al*, 2010; Di Palma *et al*, 2011; Wisniewski *et al*, 2011a; Sun *et al*, 2013; Erde *et al*, 2014; Kulak *et al*, 2014; Zougman *et al*, 2014). However, these protocols have limited flexibility due to several shortcomings, including reagent incompatibilities (detergents, chaotropes, salts), the required use of detergent alternatives (e.g., amphipols), restrictions related to absolute sample volume, throughput, and excessive handling. Subsequently, these workflows have typically been limited to processing of absolute material quantities > 1 μg or achieve reduced proteome coverage (~2,000 total proteins) when examining sub-microgram amounts of protein. These drawbacks have largely precluded the use of proteomics in applications where high reproducibility, sensitivity, and throughput are necessary, such as in clinical studies or population screening.

The rapid expansion of next-generation sequencing has prompted the development of methods amenable to high-throughput genome library preparation that are greatly facilitated by the application of paramagnetic beads in manual and roboticized

European Molecular Biology Laboratory, Genome Biology Unit, Heidelberg, Germany
*Corresponding author. Tel: +49 6221 3878560; E-mail: jeroen.krijgsveld@embl.de

platforms (DeAngelis *et al*, 1995; Wilkening *et al*, 2013). However, paramagnetic bead usage is not common in general proteomics, although they have been employed in specialized applications for covalent coupling or affinity purification of proteins, immobilized proteolysis (Fan *et al*, 2014), and for the enrichment of post-translationally modified peptides (Yeh *et al*, 2012; Zeng *et al*, 2012). Recent technologies based on nanodiamond particles have illustrated the depletion of contaminating substances and enhancement of compartment-specific proteomics (Chen *et al*, 2006; Pham *et al*, 2013). Building on technology developments pioneered by solid-phase reversible immobilization (SPRI) (DeAngelis *et al*, 1995) and nanodiamond technologies (Chen *et al*, 2006), and with the goal of enhancing and simplifying generic proteomics sample processing, we have developed SP3. SP3 is a novel single-tube proteomics workflow that provides efficient unbiased binding of proteins and peptides, enabling rapid and efficient completion of common proteomics workflows in a high-throughput manner.

In this study, SP3 is applied to a variety of conventional and ultrasensitive proteomics applications. Based on the observation that SP3 provided an enhanced platform for handling sub-microgram amounts of material determined from in-depth proteome profiling HeLa cells, we applied SP3 to examine embryonic development using *Drosophila* embryos. While a wealth of gene expression data exists for *Drosophila*, its proteome dynamics during development has been the focus of relatively few studies (Carmena, 2009). With an SP3-based approach, the proteome here is profiled to a depth of > 6,000 proteins, of the predicted 18,000 proteins in the *Drosophila* genome, from pooled embryos at 2–4 h (stages 5–7) and 10–12 h (stages 13–15) of development. This analysis was extended to capture dynamics in an ultrasensitive screen at a single-embryo resolution. These data represent the largest catalog of the *Drosophila* embryo proteome to date, while providing unparalleled sensitivity for quantitative comparisons that have the potential to reveal novel inter-individual proteome variance. Furthermore, the use of SP3 in these studies illustrates its potential advantages in other areas of developmental and clinical biology where reproducible in-depth quantitative analysis is required to explain inter-individual variation with scarce sample amounts.

## Results

Here, we demonstrate for the first time that proteins and peptides can be immobilized on the hydrophilic surface of carboxylate-coated paramagnetic beads in an unbiased fashion, initiated by the introduction of an organic additive, and by a mechanism similar to hydrophilic interaction chromatography (HILIC) (Alpert, 1990) or electrostatic repulsion hydrophilic interaction chromatography (ERLIC) (Alpert, 2008) (Fig 1A). The addition of an organic solvent to an aqueous solution containing paramagnetic beads promotes trapping of proteins and peptides in a solvation layer on the hydrophilic surface of the beads. This interaction can be adjusted through modulation of the solution pH, where an acidic solution promotes HILIC-style binding, and basic conditions are similar to ERLIC with repulsion driven by the negatively charged carboxylate group on the bead surface. We have observed this mechanism to be effective

utilizing beads from a range of manufacturers (Supplementary Fig S1A).

Once immobilized on-bead, proteins and peptides can be rinsed while on a magnetic rack (Supplementary Fig S1B and C) with a combination of solutions to efficiently remove contaminating agents, such as detergents and chaotropes. We have found that a combination of rinses with 70% ethanol and 100% acetonitrile provides optimal removal of a range of reagents common to proteomics (Fig 1B). After rinsing, proteins and peptides are eluted into an aqueous solution. At this stage, purified proteins can be directly used in a variety of downstream protocols, such as fractionation or digestion. Employing a similar workflow, peptide mixtures can be immobilized and rinsed on the surface of the paramagnetic beads. SP3 of peptide mixtures accomplishes both cleanup and concentration, eliminating the need for common de-salting and rotary evaporation steps. Subsequently, eluted peptide mixtures can be immediately subjected to MS analysis. Alternatively, peptides can be selectively eluted in a stepwise manner through modulation of the acetonitrile concentration and fractionated off-bead using ERLIC or HILIC conditions prior to MS analysis. Preceding fractionation or peptide SP3, peptides may also be chemically labeled. Each SP3 process (protein and peptide) is rapid, requiring just 15 min (excluding digestion times), and can be completed entirely in parallel with no increase in process time, even when scaling to a 96-well format. Furthermore, all steps in a conventional proteomics protocol (cell lysis, protein cleanup and digestion, peptide labeling, desalting, fractionation, and concentration) can be completed entirely in a single tube with SP3, maximizing throughput while minimizing potential sample loss.

To illustrate the efficiency and utility of SP3 in comparison with conventional methods for protein manipulation, we employed a whole-cell lysate prepared in 1% SDS-containing buffer derived from the yeast *Saccharomyces cerevisiae*. SDS–PAGE analysis of SP3-treated lysates showed no discernible protein loss when compared with untreated controls (Fig 1C). In contrast, when utilizing conditions previously described to promote protein binding to nanodiamond particles through modulation of the solution pH or bead concentration (Chen *et al*, 2006; Pham *et al*, 2013), interestingly reduced interaction with the paramagnetic beads used with SP3 was observed (Supplementary Fig S2A and B). Non-specific losses that limit protein recovery were not observed when compared to precipitation conditions as employed in SPRI for oligonucleotides, as well as a common ultrafiltration-based approach (filter-aided sample preparation, FASP) (Wisniewski *et al*, 2009, 2011b) (Supplementary Fig S2C and D). Protein enrichment with SP3 was observed to be efficient in both concentrated and dilute solutions (Supplementary Fig S3A and B), as well as in the presence of harsh sample solubilization matrices, including 10% SDS and Laemmli loading buffer, conditions incompatible with ultrafiltration membranes, and nanodiamond enrichment (Supplementary Fig S3C and D).

After proteolysis, peptide mixtures are commonly handled in a variety of downstream workflows targeted at cleanup, concentration, or labeling. These protocols use a diverse range of reagents and necessitate numerous processing steps. Simplification of these procedures while maintaining their benefits is essential for ultrasensitive proteomics. To illustrate the enhanced completion of these procedures with SP3, we employed peptide mixtures derived from a trypsin-LysC-digested yeast protein lysate. MS analysis of

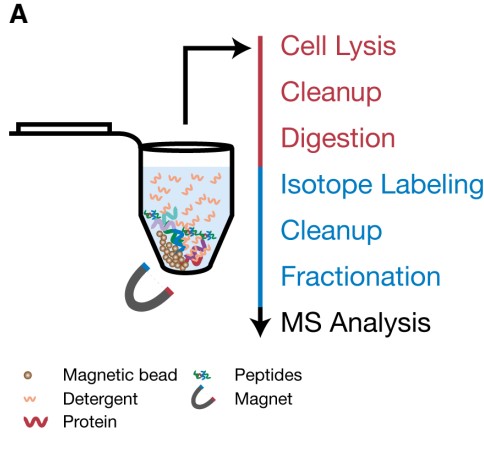

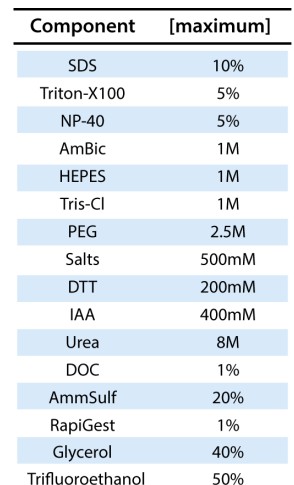

| Component | [maximum] |
|---|---|
| SDS | 10% |
| Triton-X100 | 5% |
| NP-40 | 5% |
| AmBic | 1M |
| HEPES | 1M |
| Tris-Cl | 1M |
| PEG | 2.5M |
| Salts | 500mM |
| DTT | 200mM |
| IAA | 400mM |
| Urea | 8M |
| DOC | 1% |
| AmmSulf | 20% |
| RapiGest | 1% |
| Glycerol | 40% |
| Trifluoroethanol | 50% |

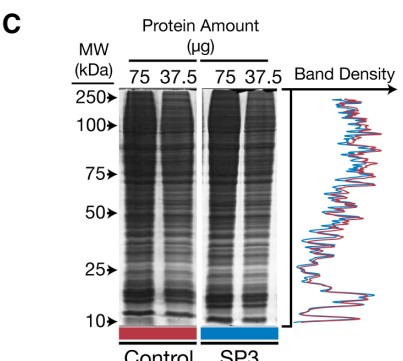

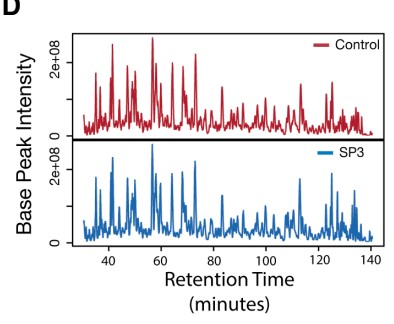

**Figure 1. SP3 provides an efficient means for preparing protein and peptide samples for MS analysis.**

A   Schematic of the SP3 workflow in a single tube. Protein and peptide mixtures are bound to carboxylate-coated paramagnetic beads through the addition of acetonitrile in a manner similar to HILIC and ERLIC. Immobilization on the bead surface permits rinsing and removal of contaminating substances prior to proteolysis or MS analysis. Elution is performed directly into aqueous solution. Red text indicates steps carried out at the protein level, and blue are performed on peptides.

B   SP3 is compatible with a variety of commonly used reagents. Table of common reagents used in proteomics studies that we have tested and determined to be compatible with SP3. Listed values are the maximum concentrations tested. Reagents that do not appear in this table have not been tested and may be compatible with SP3.

C   SP3 demonstrates high recovery for both proteins and peptides. SDS–PAGE analysis of a yeast whole-cell lysate left untreated (Control) or treated with SP3. Numerical values at the top of each lane indicate the amount of starting material (μg of protein). Plot on the right displays overlaid densitometry data from the 37.5 μg lanes.

D   Base-peak chromatograms of equivalent peptide mixtures analyzed by MS after treatment with StageTips (Control) or with SP3.

SP3-treated peptide mixtures showed no apparent peptide loss when compared with control samples prepared using a conventional StageTip procedure (Rappsilber *et al*, 2003) (Fig 1D). Improved peptide recovery was observable when comparing SP3 conditions to pH, high bead concentration, or precipitation conditions utilized in nanodiamond and SPRI approaches (Supplementary Fig S4A–C). In addition, there was no discernible difference between chromatograms from MS analysis of replicate enrichments (Supplementary Fig S4D), indicating that these benefits did not come at the cost of reproducibility.

Prior to, or as part of, cleanup and concentration procedures, it is common to chemically label peptide mixtures with stable isotopes to facilitate quantitative comparisons between treatment and control conditions. The two most frequently applied chemical labeling approaches, reductive dimethylation (Boersema *et al*, 2009) and isobaric mass tagging (Thompson *et al*, 2003; Ross *et al*, 2004), are based on amine-reactive chemistry and thus have specific buffer and solution composition requirements. Using a peptide mixture derived from a yeast whole-cell lysate, along with a simple modification to the SP3 protocol (Supplementary Fig S5A), > 98% efficiency of labeling with both dimethyl and tandem mass tags (TMT) (Fig 2A) was observed. The reproducibility of the labeling and enrichment reactions was found to be high as illustrated by the minimal deviation from a fold change of zero for peptides in TMT 6-plex experiments of biological replicate samples (Supplementary Fig S5B). Biological reproducibility was also found to be high even when handling sub-microgram amounts of material in an SP3-based single-tube workflow. This was demonstrated by the high reproducibility in protein quantification (mean Pearson correlation: 0.89) (Fig 2B) when comparing mouse embryonic stem cells (mESC) and neural progenitor (NP) cells in 10 biological replicates, each starting from 5,000 cells prepared in 1% SDS-containing buffer as input. Together, these data indicate that SP3 provides a means for simple and efficient completion of a variety of established quantitative proteomics workflows.

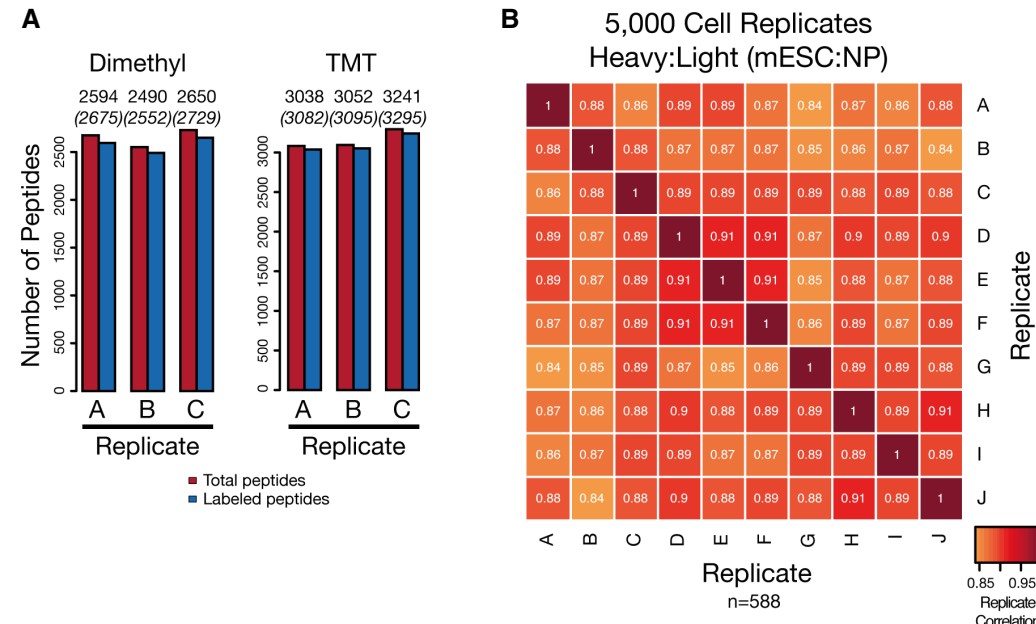

**Figure 2.  SP3 facilitates efficient and reproducible chemical isotope labeling of peptide samples.**

A    SP3 promotes efficient labeling of peptide mixtures. Labeling efficiency of dimethyl and TMT methods when coupled to SP3 as measured by the number of peptides identified as fully labeled or partially/not labeled with the expected tag in triplicate measurements. Values above columns indicate numbers of identified (in brackets) and labeled peptides.

B    SP3 enhances quantification reproducibility in quantity-limited samples. Replicates of 5,000 cell populations of mESC and NP cells were prepared using SP3 and analyzed with single-shot injections and a dimethyl tagging approach for quantification. A total of 10 individual biological replicates for each cell type (20 total samples) were prepared and analyzed. Intensity of each box represents the Pearson correlation between the heavy:light transformed (VSN) peptide areas translated into protein ratios (*n* = 588), with values displayed as text. Peptides were required to be quantified as heavy:light pairs in a minimum of 9 biological replicates.

A significant amount of effort in proteomics is focused on achieving in-depth proteome coverage both in abundant and in quantity-limited applications. To assess performance with in-depth proteome profiling in abundant samples, we analyzed a yeast whole-cell lysate (~10 μg of protein from a lysis prepared in 1% SDS-containing buffer) treated with SP3 or filter-aided sample preparation (FASP) prior to high-pH reversed-phase fractionation. Combining biological duplicates, comparable coverage between SP3 and FASP was observed, with a total of 3,944 and 4,008 proteins identified (Supplementary Fig S6A), mapping to 39,211 and 43,318 unique peptides, respectively. This identity extended to reproducibility between individual replicates (Supplementary Fig S6A), protein abundance distributions (Supplementary Fig S6B), and intensity-based absolute quantification (iBAQ) (Schwanhäusser *et al*, 2011) assignments (Supplementary Fig S6C). Moreover, there was no discernible bias in properties of the peptides captured with each protocol, as indicated by charge state (Supplementary Fig S7A), molecular mass (Supplementary Fig S7B), isoelectric point (Supplementary Fig S7C), grand average hydropathy (GRAVY) (Kyte & Doolittle, 1982) distributions (Supplementary Fig S7D), and amino acid content (Supplementary Fig S7E). Interestingly, although we observed an expected recovery of ~50% of starting material in FASP compared with SP3 (Supplementary Fig S2C) (Wisniewski *et al*, 2011b), this did not adversely affect the depth of proteome coverage. This indicates that the differences in recovery between the protocols are indeed non-specific in nature and that the amount of starting material was sufficient to overcome these losses. Therefore, it can be concluded that SP3 has no observable bias and is compatible with total-proteome

characterization when compared with a widely employed method, FASP.

One of the primary strengths of the SP3 approach is the ability to provide scalable protein and peptide recovery in a single-tube work-flow, maximizing potential efficiency for ultrasensitive applications. Previous attempts at analyzing small amounts of material have faced challenges during sample processing and have frequently relied on the preparation of large cell populations diluted to working concentrations (typically in the range of approximately 1,000–5,000 cells) after lysis and digestion (Altelaar & Heck, 2012). Instead, here we assessed the performance of SP3 for in-depth proteome analysis starting from sub-microgram amounts of material. Specifically, replicate HeLa cell lysates from populations totaling 500,000, 50,000, 5,000, and 1,000 cells were prepared using a simplified detergent-based lysis coupled to the single-tube SP3 protocol.

After fractionation with high-pH reversed-phase HPLC, high-pH reversed-phase StageTip, SP3 fractionation (ERLIC-style), or single-shot injections, efficient protein and peptide recovery was observed across a wide quantity range determined by base-peak chromato-gram complexities and intensities (Fig 3A). This resulted in high depth of proteome coverage based on unique peptide identifications in single-shot and fraction injections of low abundance samples (Fig 3B). Strikingly, even when starting from 1,000 cells, we reliably identified > 15,000 unique peptides (Fig 3B). However, the 1,000 cell samples approached the limit in sensitivity of the MS hardware, resulting in a loss of total proteome coverage and sample intensity, and subsequently in a skewed distribution of iBAQ values (Supplementary Fig S8A). In contrast, there was no observable difference in

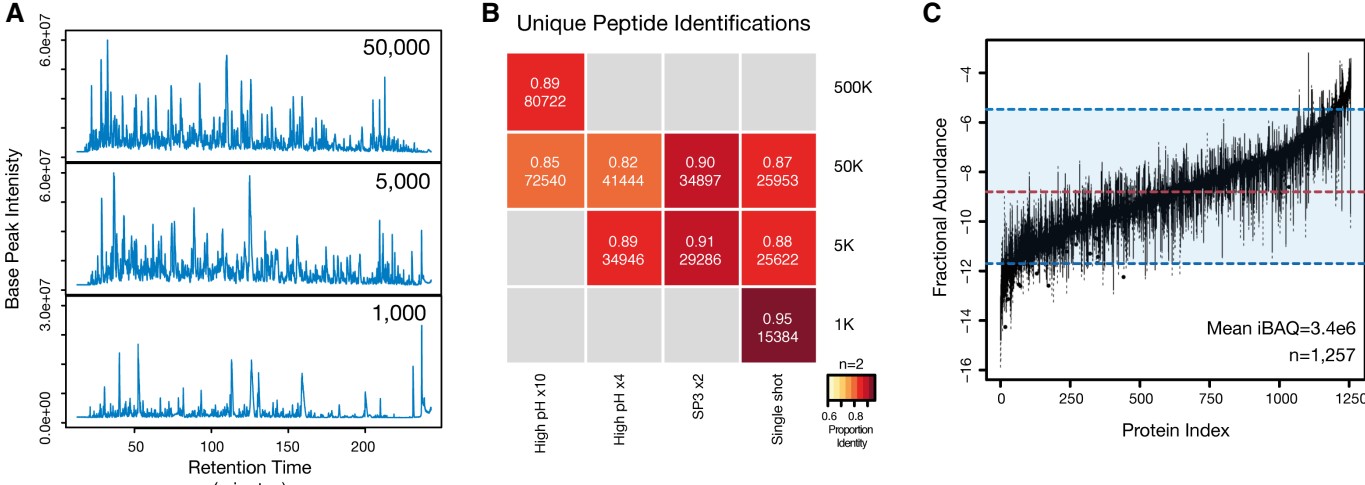

**Figure 3. SP3 is compatible with and enhances ultrasensitive proteome analysis.**

A SP3 facilitates enhanced depth of coverage in quantity-limited samples. Base-peak chromatograms from single-shot injections of peptide mixtures prepared using a single-tube SP3 protocol from 50,000, 5,000, and 1,000 HeLa cells. In the cases of the 5,000 and 1,000 samples, the entire recovered peptide amount was injected. For the 50,000 cell sample, 25% of the recovered amount was injected to avoid overloading of the chromatography column.

B Number and reproducibility of unique peptides (by sequence) identified in each analysis. The intensity of color in each block denotes the percentage overlap in peptides between replicate samples ($n$ = 2 for all samples). Values in each block denote the fraction overlap and the total number of unique identifications from combined biological duplicates. The number following the × in each method indicates the number of fractions analyzed.

C SP3 enables proteome profiling across a wide range of proteome abundance in quantity-limited samples. Box plot of fractional protein abundance values estimated from iBAQ values from the 50,000, 5,000, and 1,000 single-shot injections. Values for each protein represent mean iBAQ values from combined biological duplicates. Only proteins identified in all three combined single-shot samples are used in the analysis ($n$ = 1,257). Dashed red and blue lines indicate the median and a range spanning 5–95% of log (iBAQ) values.

the abundance profiles (Fig 3C, Supplementary Fig S8A) between the 50,000 (25% injected) and 5,000 cell single-shot samples, likely stemming from saturation in the instrument sampling rate (Supplementary Fig S8B). This highlights the similarity of the captured proteomes in contrast to the difference in starting material.

Despite prior developments in protocols aimed at the preparation of ultrasensitive samples, none is yet accepted as a 'gold standard' for proteomics. Although FASP has seen widespread utility for sample processing, the requirement for modifications to overcome losses associated with the method has limited its utility in ultrasensitive screens (Wisniewski *et al*, 2011a,b; Erde *et al*, 2014). To benchmark SP3 in relation to current state-of-the-art high-efficiency proteomics methods, we compared with the data as presented by the recently described integrated Stage-Tip (iST) protocol (Kulak *et al*, 2014). Although not as flexible as FASP in terms of reagents, iST boasts excellent performance in handling of minute quantities of material. In the original paper, this is highlighted in an ultrasensitive analysis of limited numbers of HeLa cells (Supplementary Fig S1D in Kulak *et al*). Comparing the numbers of identified peptides, the depth of coverage obtained with 10,000 cells by the iST protocol requires just 5,000 cells with the SP3 approach. Interestingly, we observed an increase of ~5,000 unique peptide identifications when analyzing equivalent cell numbers (1,000 cells) with SP3 compared to the iST data. It is noteworthy that although both protocols focus on the minimization of processing volumes and encapsulation of all steps in a single device, the reversed-phase nature of the iST method imparts the limitations of conventional proteomics methods on the technology (e.g., inability to use detergents such as SDS or solvents such as trifluoroethanol (TFE) during lysis). In contrast, SP3 does

not suffer from any of the limitations that hinder ultrasensitive proteomics methods, and as a result, we were able to utilize 1% SDS to enhance cell lysis and protein solubilization, potentially explaining the increased numbers of identified peptides.

To capitalize on the benefits of the SP3 approach, it was applied in a whole-proteome analysis of *Drosophila* embryos at 2–4 and 10–12 h after egg lay (AEL). Previous proteomics analyses of *Drosophila* embryos have necessitated the use of milligrams of material, equating to hundreds of individual embryos combined across multiple stages of embryogenesis (Brunner *et al*, 2007; Zhai & Villén, 2008; Gouw *et al*, 2009). This experimental design limits the resolution of the analysis by eliminating stage-specific developmental dynamics as well as averaging out variability between individuals. We applied a modified SDS-TFE-lysis protocol designed to eliminate the need for physical disruption (e.g., bead milling or beating, Dounce homogenization), in combination with a single-tube SP3 workflow to 60 embryos, and identified a total of 5,632 unique gene products, including prominent developmental factors, such as *bicoid*, *hunchback*, and *nanos* (harvested between 2 and 4 h AEL, 12 MS runs, $n$ = 1) (Fig 4A, Supplementary Table S1). Previous studies have estimated the number of polyadenylated RNA species to be between 5,000 and 7,000 (Levy & Manning, 1981), and the number of gene products in the range of 8,000–10,000 in 2- to 4- and 10- to 12-h stages (Graveley *et al*, 2011). Thus, the SP3-generated proteome covers > 60% of the predicted gene models. These data represent a significant increase in depth of coverage compared with a previous benchmark study utilizing hundreds of embryos from multiple stages (2- or 24-h pooled AEL, 3,856 genes) and 62 individual MS runs (Brunner *et al*, 2007).

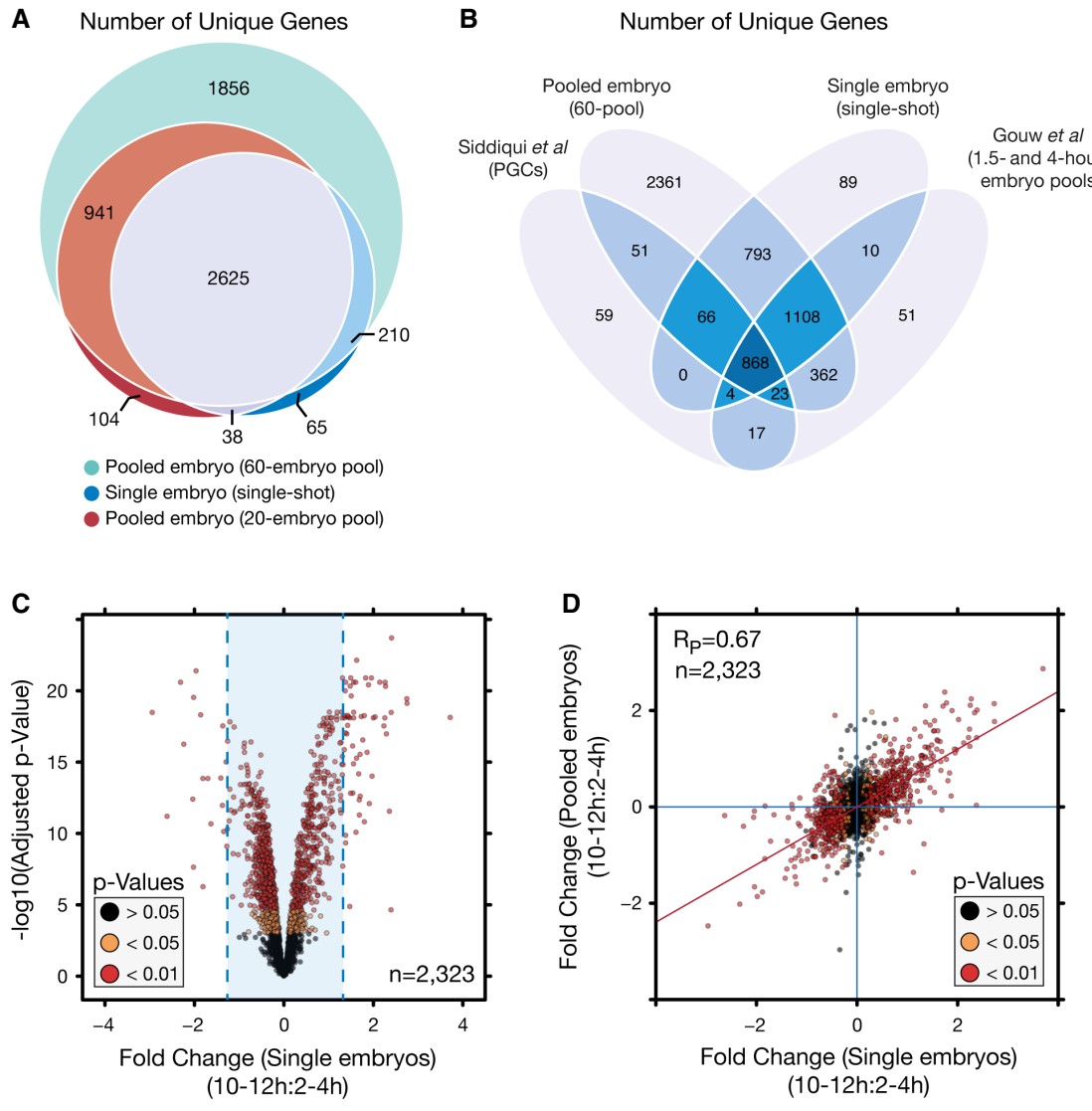

**Figure 4.  SP3 facilitates enhanced qualitative and quantitative analysis of single *Drosophila* embryos.**

A   In-depth proteome coverage can be obtained across a wide range of pool sizes down to the single-embryo level with SP3. Venn diagram depicting the number of unique gene products (FlyBase accession) identified between different starting pools of embryos (60-embryo pool, $n = 1$, 20-embryo pool, $n = 2$, single embryos, $n = 11$). All samples are combined identifications from 2–4 to 10–12 h samples.

B   Number of unique gene products identified between single-embryo samples and published datasets focused on staged developmental proteome analysis.

C   SP3 permits quantitative analysis at the single-embryo level. Volcano plot depicting protein variance between 2- to 4-h and 10- to 12-h developmental time points in single-embryo samples. Fold changes were determined as a trimmed mean of VSN transformed peptide values. *P*-values were determined using limma with Benjamini–Hochberg correction for multiple testing. Values were calculated across a total of 11 biological replicates. Blue lines indicated the mean fold change, $0.014 \pm 2.5$ times the standard deviation.

D   Scatter plot of protein fold-change values between pooled and single-embryo samples illustrating the limited variation between fold-change values determined between the two study designs. Colorization is based on *P*-values determined through comparison of 2- to 4-h and 10- to 12-h stages in single-embryo samples. Blue lines indicate zero-fold-change values, and the red line is a linear fit to the data.

To examine proteome dynamics between developmental stages, a dimethyl tagging approach coupled with SP3 was utilized (Supplementary Fig S9A). With combined replicates ($n = 2$), a total of 3,870 and 3,881 unique gene products (3,892 combined) could be identified in 2- to 4- and 10- to 12-h staged samples (Fig 4A, Supplementary Fig S9B, Supplementary Tables S1 and S2). This depth of coverage required just 20 pooled embryos per stage and, in spite of the limited amount of starting material, provided excellent quantitative reproducibility prior to (Supplementary Fig S9C)

and following data normalization (Supplementary Fig S9D), as well as between replicates (Supplementary Fig S9E). Thus, the use of SP3 enabled in-depth analysis of a limited sample to achieve quantifiable dynamics of the developmental proteome from pooled embryos.

Recent work with *Xenopus laevis* has illustrated the quantitative analysis of expression kinetics at a single-embryo resolution (Sun *et al*, 2014). This was facilitated by the large size of the *Xenopus* embryo (> 1.2 mm) coupled to pooling with multiplexed isobaric

tagging. Examination at a single-embryo resolution affords investigation of inter-individual variability not possible in pooled samples. To achieve this resolution in significantly smaller (< 500 µm) individual *Drosophila* embryos, a single-tube SP3 protocol was utilized. Due to the limited amount of material, efficient lysis and capture of the resultant material was essential. The flexibility of SP3 allowed for the use of the harsh solution-based SDS-TFE protocol in combination with sonication to provide efficient and reproducible lysis, and eliminated the need for physical disruption methods (e.g., bead beating) that would result in unacceptable material losses when handling a single embryo. From a collection of 11 biological replicates per stage (22 total samples), coverage of a total of 2,938 unique genes was obtained (97.8% of genes covered in pooled samples) (Fig 4A). While we estimate a single embryo contains just ~200 ng of protein, significant complexity is observable within each single-shot analysis (Supplementary Fig S10A), leading to a high sampling rate (Supplementary Fig S10B).

Although a portion of the embryo proteome is missed due to the low quantity of starting protein (Fig 4A), the similarity in depth of coverage of identified gene products compared with previous stage-specific studies (Fig 4B) utilizing considerably larger amounts of material indicates that SP3-based single-embryo analyses provide sufficient information for comparative experiments. Quantitatively, the data revealed few differences from the pooled samples, where averaging of embryos would be expected to minimize variation (Supplementary Table S2). Nominal deviations in reproducibility (single: 0.69, pooled: 0.74, Pearson correlations) (Supplementary Fig S10C) and in mean fold change (single: 0.014 SD ± 0.31, pooled: 0.018 SD ± 0.15) (Supplementary Fig S10D) between replicates and stages indicated the quality of the quantitative information relative to the pooled samples. A total of 17 and 59 proteins meeting a minimum *P*-value cutoff of 0.05 (Benjamini–Hochberg correction) and an absolute fold change > 2.5 times the standard deviation (Fig 4C, Supplementary Table S2) were observed as enriched between the 2- to 4- and 10- to 12-h stages.

Importantly, the directional trends in protein expression are highly correlated (Pearson correlation = 0.67, *n* = 2,323) between the pooled and single-embryo data (Fig 4D), further highlighting the ability to extract reliable quantitative information from single embryos.

Examination of functional annotation within these sets revealed genes involved in a range of cellular processes (Supplementary Fig S11A). Genes associated with mitosis and meiosis (*polo*, *Klp3A*, *BRWD3*), stress response (*ref(2)p*, *homer*), or chromatin and chromosome organization (*aur*, *CG3509*) were enriched in 2–4 h collections. A significant number of genes associated with neural development (*Fas2*, *betaTub60D*, *Nrg*, *Nrt*, *fax*, *Ama*, and *hts*) were abundant in 10- to 12-h pools. In addition to neural processes, proteins involved in splicing (*U2af50*, *Pep*), chromosome structure or function (*Dsp1*, *D1*, *mod*), and regulation of transcription (*Ref1*, *pzg*) were enriched. In both stages, new proteins whose biological functions were uncharacterized were identified as enriched. Interestingly, comparison of the top 20 increased proteins in each stage with gene expression data (Graveley *et al*, 2011) revealed that many of the observed abundance changes were also detected in RNAseq (Supplementary Fig S11B) and RNA *in situ* hybridization (Supplementary Fig S12) experiments targeting the equivalent points in *Drosophila* development, further validating the directionality of the observed trends.

In addition to general abundance, previous stage-specific analyses of primordial germ cells and staged embryo pools have revealed diverse proteome alterations during the early and late phases of the maternal-to-zygotic (MZT) transition (Gouw *et al*, 2009; Siddiqui *et al*, 2012). The bulk of zygotic transcription initiates at ~2 h post-fertilization (Tadros & Lipshitz, 2009), and as such, the levels of the associated proteins would be expected to rise. Using associated proteins identified in the single-embryo data (Fig 5A), a significant difference in zygotic and maternal associated protein expression was observed (*P* = 3.43e-21, Mann–Whitney *U*-test, zygote *n* = 251, maternal *n* = 139) (Fig 5B). Zygotic genes previously found to increase during MZT (Gouw *et al*, 2009), such as *bnb* and *ama*,

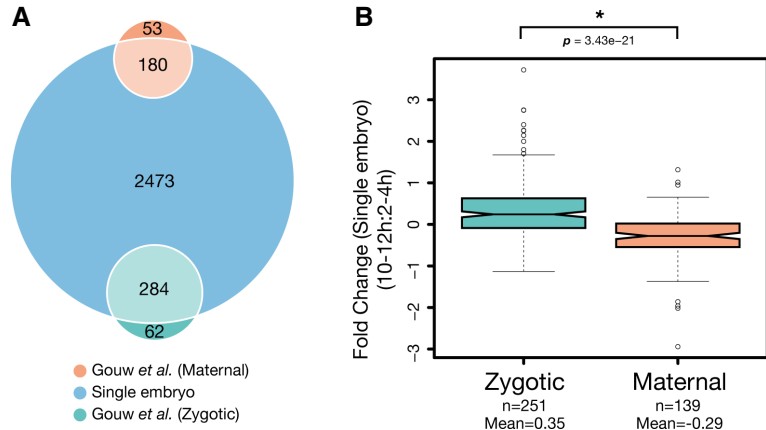

**Figure 5.  Increased zygotic-associated expression is observed in 10–12 h embryos relative to those from 2 to 4 h.**

A   Venn diagram depicting the number of identified maternal and zygote-associated gene products in this study. Single-embryo data were annotated and compared with previously determined maternal-to-zygotic expression data (Gouw *et al*, 2009). Comparisons were made based on FlyBase gene accessions from both datasets.

B   Notched box plot of fold-change values from maternal and zygote-associated gene products between 2- to 4-h and 10- to 12-h samples. Whiskers indicate 1.5× the interquartile range, plus or minus the values for the third or first quartiles, respectively. *P*-values were determined using a Mann–Whitney *U*-test.

were among those with the highest rise in abundance, whereas maternal genes like *yl* were significantly decreased in 10- to 12-h samples relative to 2–4. From a set of 27 genes classified as having strictly maternal expression (Arbeitman *et al*, 2002), 11 are identified in this study. Only 6 (*me31B, DNApol-alpha50, CG3800, wech, Pxt*, and *Cyt-b5*) are contained in the final quantification dataset, with the other 5 (*smg, bru-2, CG7627, CG5568*, and *fs(1)M3*) falling below our stringent threshold for reliable detection across replicates due to their low abundance. The remaining 6 have a mean fold change of −0.53, emphasizing their low abundance in the later stages of development.

In addition to variation in protein abundance between developmental stages, examination at the single-embryo level facilitates inter-individual analyses not feasible in pooled samples. To examine the ability to measure differences between individuals, standard expression maps for 2- to 4- and 10- to 12-h time windows were prepared using pooled embryos (*n* = 2 per stage, 20 embryos per pool). Single embryos were then analyzed, and the correlation between protein fold-change values used to determine similarity with the standard expression maps. All fold-change values were relative to a common internal standard (pooled embryos from all compared stages) to facilitate comparison between samples. We focused on proteins that were detected in all samples (*n* = 1,019) and then examined quantitative differences in their abundance between individual embryos.

These data revealed significant differences in abundance for proteins quantified from single embryos collected in 2- to 4- and 10- to 12-h time windows (Supplementary Fig S13A and B, Supplementary Table S3). The expected difference in protein expression between the two developmental stages could be easily recognized in single embryos and, in the case of 10–12 h, can even be used to map them on a developmental timescale based on their shared similarity with the corresponding pool (Fig 6A, upper triangular). When this proteome set is narrowed to a subset of proteins that show differential expression (determined as the maximum difference in fold change that generated a sufficient pool of candidates for a reliable correlation) between the two stages, the clarity of the mapping could be improved (Fig 6A, lower triangular). Unsurprisingly, this subset of 25 proteins contained numerous proteins, such as *bnb, Bacc, fax, Cys, Nrg, Fas2,* and *yl* previously observed to differ in a stage-specific manner at both the transcriptional and translational levels in the single-embryo screen.

To determine whether abundance patterns could be used to map embryos where total proteome differences are expected to be small, standard expression maps were prepared using 20 embryo pools (*n* = 2 for each stage) from 2- to 2.5- to 3.5- to 4-h time windows. Single embryos collected in a 2- to 4-h time window were then analyzed to determine similarity with the expression maps and to allow more precise molecular staging of each individual. Based on the quantified proteins from the single embryos, few significant

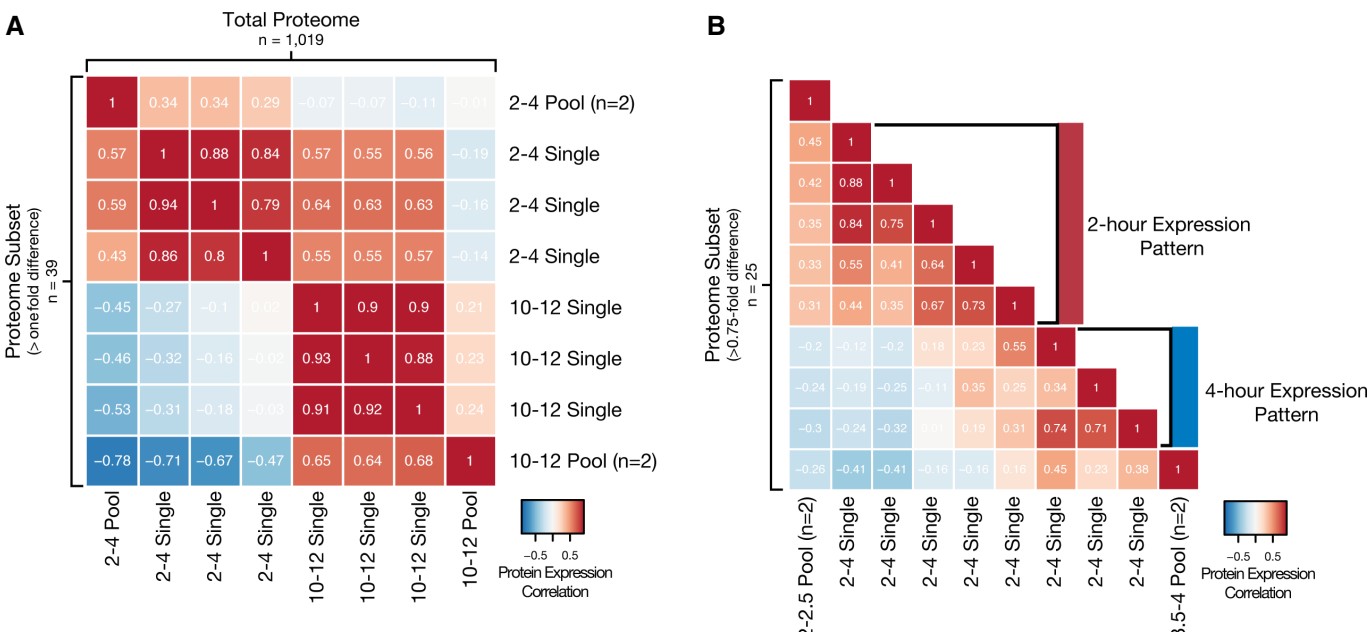

**Figure 6.    The quantitative resolution provided by SP3 permits tracing of embryo origin based on protein expression patterns.**

A    Single embryos can be mapped between time points with large divergence in protein expression. Low-resolution expression mapping based on 2- to 4-h and 10- to 12-h developmental stages. Heat map depicts the correlation between fold-change values relative to an internal standard for all proteins quantified (*n* = 1,019, upper triangular) or those with a difference in fold-change value > 1 (*n* = 39, lower triangular) between pooled (20 embryos each, *n* = 2 for each stage) and single embryos (*n* = 3 for each stage).

B    Single embryos can be mapped between time points with minimal divergence in protein expression. High-resolution expression mapping based on 2- to 2.5-h and 3.5- to 4-h developmental stages. Heat map depicts the correlation between fold-change values relative to an internal standard for proteins quantified with a difference in fold-change value > 0.75 (*n* = 25) between pooled (20 embryos each, *n* = 2 for each stage) and single embryos (*n* = 8).

Data information: Color intensity is based on the Pearson correlation with values displayed in boxes.

differences in fold change were detectable due to the similarity between the stages (Supplementary Fig S13C and D, Supplementary Table S4). However, when the dataset was narrowed to include only those proteins that show differential expression (determined as the maximum difference in fold change that generated a sufficient number of candidates for a reliable correlation), clusters of individual embryos that displayed expression profiles similar to 2–2.5 and 3–3.5 h samples could be discerned (Fig 6B).

To further validate the expression patterns of the subset of candidate proteins, we utilized high-resolution expression data from sectioned embryos (Combs & Eisen, 2013). Expression data were derived from 25-μm sections of embryonic stage 2 (25–65 min AEL), 4 (80–130 min AEL), and 5 time points within stage 5 (130–170 min AEL). For a majority of the 25 candidate proteins, the corresponding expression pattern could also be detected at the RNA level (Supplementary Fig S13E). Of these 25 proteins, 10 were associated with the maternal-to-zygotic transition. This developmental process directly overlaps with the 2- to 4-h collection window. From these 10 proteins, nine had expression profiles available in the section data. A total of eight of these exhibited the expected directionality in expression also observed in the protein data based on knowledge of the maternal-to-zygotic transition (Supplementary Fig S13F, maternal: CG3663, iPLA2-VIA; zygotic: Nrt, bnb). Together, these data highlight the ability to detect differences at a high resolution that can be validated with the knowledge of cellular processes or expression dynamics.

## Discussion

In this work, we have presented a novel concept for proteomics that simplifies and streamlines a range of conventional handling procedures common to MS-based protein analysis experiments. SP3 affords efficient completion of numerous proteomics protocols combining all steps from cell lysis to fractionated peptide samples ready for MS analysis in a single-tube workflow. As such, SP3 offers several advantages compared with current state-of-the-art proteomics technologies in the areas of flexibility, scalability, and throughput. Importantly, SP3 combines these benefits into a single platform, effectively simplifying sample preparation and eliminating the need for significant protocol optimization prior to proteome analysis. The data presented in this work demonstrate the feasibility of performing ultrasensitive, high-throughput analyses of challenging biological specimens with the SP3 approach, opening new avenues of research to the proteomics community.

The diverse nature of the proteome necessitates the use of an equally broad set of reagents for its preparation. This is especially important when handling difficult matrices, such as tissues, where harsh denaturants are essential. The flexibility and benefits of using these reagents are demonstrated throughout this work, where all experiments in yeast, human, mouse, and *Drosophila* were performed in the presence of 1% (or greater) concentrations of SDS. As a result, SP3 provides unmatched flexibility in catering reagent conditions to enhance analysis of protein subsets (e.g., membrane) or total proteome mixtures. Although protocols such as FASP offer the ability to use a broad set of reagents, this is limited by the compatibility of the filter unit membrane. An added advantage of SP3 is that these benefits extend to the peptide level, where

proteolysis-enhancing reagents (e.g., urea, deoxycholate) can be used without protocol modification due to their removal during SP3 treatment. This circumvents the need for tedious acid precipitation or phase-transfer protocols for detergent removal, or dilution and desalting steps in the case of urea.

Notably, the flexibility of the SP3 protocol also applies and was proven to be essential to ultrasensitive applications in this study. Although mammalian cells can be easily lysed with SDS treatment, yeast and embryo samples present a significant challenge and are conventionally tackled using physical disruption methods (e.g., bead beating, Dounce homogenization). In the *Drosophila* embryo work data presented in this study, single-embryo screening required the use of a high-efficiency, lossless lysis protocol that could not rely on disruption with physical means due to the limited sample quantity and the inefficiency of these methods. In our hands, effective and reproducible lysis could only be obtained with a combination of detergents and organic solvents. This reagent combination renders the lysate incompatible with numerous processing protocols, including the recently published, state-of-the-art iST approach (Kulak *et al*, 2014).

The extension of this harsh treatment method to an ultrasensitive application further highlights the robustness of SP3. Due to the losses associated with the removal of contaminants (e.g., detergents used to aid in unbiased and complete protein solubilization), they are typically excluded in ultrasensitive processing methods. Although acid-labile detergents and amphipols (Ning *et al*, 2013) offer an attractive alternative to conventional detergents, they nevertheless require removal with acid precipitation or phase transfer prior to MS analysis. These extra processing steps increase the potential for protein and peptide loss and limit the sensitivity of low-quantity samples. The ability of SP3 to scale from pools of 60 to single embryos without method adaptation renders it unique in this aspect among proteomics protocols and facilitated acquisition of an entirely novel dataset targeting *Drosophila* embryogenesis.

In addition to challenges concerning protein extraction and processing, SP3 provided essential enhancements for quantitative handling of proteins and peptides. As efficient extraction from a single embryo yields only an estimated 200 ng of protein, scalable and lossless processing is vital. The reproducibility of these extraction and processing steps is paramount for minimizing technical noise that can dominate inter-sample signal when working with limited quantities of material. SP3 exhibited high performance in all of these critical areas, demonstrated by in-depth profiling of HeLa samples originating from as few as 1,000 cells, and unmatched depth of coverage in pooled and single embryos. Impressively, the results in the HeLa analyses were obtained from samples starting from the indicated low cell numbers, which is in contrast with the common practice of examining a diluted sample of material obtained from a larger pooled lysate derived from millions of cells (Altelaar & Heck, 2012). Preparation of samples in this manner can mask variability and losses introduced during lysis and early handling steps. By working directly with minute cell numbers, the results presented here are directly extendable to true experimental situations where 1,000 cells may be the entire available quantity. Adaptation of SP3 to microfluidic platforms where losses to plastic ware are eliminated can potentially decrease this workable cell number further.

With the ever-increasing demand for methods compatible with the analysis of large sample cohorts in tissue libraries, or

populations of genetically distinct individuals, coupling the flexibility and sensitivity of SP3 to high-throughput situations would be of significant advantage. Although SDS-Gels (Schmidt, 2013), FASP (Yu *et al*, 2014), in-solution, and iST (Kulak *et al*, 2014) methods are theoretically scalable to 96-well format, these protocols can require specialized equipment (iST) and lengthy processing (FASP) or perform poorly with small quantities of sample (FASP, SDS-Gels). The *Drosophila* single-embryo data presented in this study directly benefitted from an SP3 method that combined flexibility, sensitivity, and throughput. In addition to the rapid completion of SP3 (15 min for protein or peptide enrichment), single embryos were processed in parallel in a 96-well format. This facilitated processing a large number of individual embryos for analysis, affording an increase in statistical power. Importantly, this high-throughput processing did not result in reduced reproducibility, even when handling limited sample quantities (single embryos, mESCs, NP cells) and coupled to a chemical isotope labeling protocol (reductive dimethylation). This minimization of variance promoted examination of true biological divergence between the developmental stages examined.

Exploiting these advantages, we examined proteome dynamics during *Drosophila* embryogenesis. Using just 60 embryos of starting material, we annotated > 60% of gene products that are potentially expressed at any given stage of development. This represents the greatest depth of profiling of the *Drosophila* proteome during development and is a significant step toward proteome annotation in this fundamental model organism. Importantly, this depth of coverage was achieved using a significantly reduced number of embryos, eliminating the need to harvest hundreds to perform in-depth proteome profiling. This sensitivity afforded completion of an entirely novel screen of proteome dynamics in single embryos. By completing differential proteome analysis at this resolution, we have developed a new paradigm with which *Drosophila* screens can be completed. Despite the low sample input, the SP3 approach successfully met or exceeded proteome coverage of previous works without sacrificing quantitative accuracy or resolution between individuals. The ability to probe the proteome to a significant depth from small pools or individual embryos opens new avenues for proteomics research in *Drosophila*, where sampling limitations have previously placed restrictions on proteome examination with MS.

In addition to the identifications, the quantitative stage-specific data revealed significant diversity in protein abundance between embryos collected in 2- to 4- and 10- to 12-h (AEL) collection windows. Due to the early developmental stage, we identify proteins with increased abundance in the 2- to 4-h windows that are involved in the formation and maintenance of the embryos, such as *yl* (Schonbaum *et al*, 2000) and *ptx* (Tootle *et al*, 2011) and in the progression of early embryo development, such as *polo*. Polo is a multifunction kinase whose activity is required during mitosis, and thus, its expression is highest in regions where cells are proliferating. Mutants of *polo* have been identified through maternal effect screens and illustrate the necessity for its partial activity prior to and after the transition to zygotic expression (Glover, 2005). We also capture the maternally deposited, actin-binding protein Homer. Expression of *homer* has been described to have a redundant function to *bifocal* in maintaining posterior localization of *osk* gene products, a critical process for embryonic patterning in the early embryo (Babu *et al*, 2004). Another posterior localized protein, Vasa, was also identified as enriched in the 2- to 4-h stage. In the early

embryo, expression of *vas* has been linked to the formation of the posterior germ plasm that will eventually lead to the formation of germ cells (Mahowald, 2001). The enrichment of proteins with characterized roles in early development highlights the ability of the SP3 method to reveal protein dynamics at a single-embryo resolution.

From embryos in the 10- to 12-h stage, we observed increased abundance for proteins associated with a wide array of cellular processes. In this candidate list, we identify prominent regulatory proteins, such as Putzig, that plays a role in regulating the expression of Notch, an important factor for controlling proliferation during development (Kugler & Nagel, 2007). In *C. elegans*, the Ref-1 (Aly) proteins also identified here have similarly been implicated in Notch signaling during embryo patterning (Neves & Priess, 2005). We also observed enrichment of the protein product of the *glo* gene that has been shown to regulate *nanos* expression, an essential factor in early development during embryo patterning (Kalifa *et al*, 2006). Although the RNA expression profiles of patterning factors like *glo* extend across multiple developmental stages, the differences in their protein levels suggest potential post-transcriptional regulation. Future investigation into post-translational patterns of modifications can potentially aid in determining the relation between the transcriptional and translational status of these proteins.

In addition to proteins regulating transcription and translation, a substantial fraction of the proteins enriched in 10- to 12-h collections were associated with pathways related to neural development. Specifically, we identified many proteins with roles in axon targeting and development of the central nervous system (CNS) (*Fas2, Ama, hts, Nrg, Nrt, pod-1, fax, lamins*). Interestingly, the products of *Fas2, Nrt, Ama,* and *Nrg* all function as cell-adhesion molecules (CAMs) on the surface of axons. *Nrt* and *Ama* have been described to function together in a junction complex to regulate axon guidance (Zeev-Ben-Mordehai *et al*, 2009). *Ama* and *fax* have also been described as dominant enhancers that augment the mutant phenotype of Abelson tyrosine kinase (Abl), promoting disruption of the central nervous system (Hill *et al*, 1995; Liebl, 2003). *Fas2* and *Nrg* have both been implicated in the regulation of epidermal growth factor receptor (EGFR) signaling, the expression of which is essential for cell differentiation and development (Islam, 2003; Mao & Freeman, 2009). The products of the *hts* and *pod-1* genes have both been described to play a role in axon guidance, operating through cytoskeletal or trans-membrane associations (Rothenberg *et al*, 2003; Ohler *et al*, 2011). Lastly, the Jupiter protein also has a potential role in these developmental processes, suggested by its high, localized expression in axons (Karpova *et al*, 2006). Development of the neural system begins at ~4.5 h with the migration of neural precursor cells to the interior of the embryo (Scott & O'Farrell, 1986). Correspondingly, the majority of these candidate genes exhibit high gene expression during this transition (Graveley *et al*, 2011), consistent with the trends in protein abundance observed here.

In addition to cellular regulation, these data could be associated with the patterns of protein expression related to developmental processes. Shifts in candidate expression associated with the MZT based on transcript (Arbeitman *et al*, 2002) or protein (Gouw *et al*, 2009) data were easily distinguishable at the single-embryo level between stages. Interestingly, this diversity could also be observed between individuals collected within a single stage (2–4 h) for genes such as *bnb, Nrt, yl,* and *Nplp2*. In addition, we also observed

differential abundance in proteins such as Kugelkern, which is known to be present throughout early embryogenesis, switching from a maternally deposited isoform to a zygotic form thereafter (Guilgur *et al*, 2014). Therefore, even in this small developmental window, biologically valid abundance diversity relating to cellular shifts can be captured.

Aside from those involved in the MZT, the majority of the candidate proteins identified in the inter-individual screens exhibit high expression in the early embryo in RNAseq experiments (Graveley *et al*, 2011). Although the biological functions of many of the detected candidates are as yet uncharacterized, a variety of proteins implicated in cellular developmental processes, such as neurogenesis, are identified. In combination with histones themselves, we observed differential expression of the histone splicing and uptake regulatory protein, Slbp. Slbp is known to regulate the progression through S-phase in the cell cycle, and the increasing expression profile observed here fits with the requirement for Slbp in early development (Lanzotti & Kupsco, 2004). We also observed dynamic abundance in the essential transcriptional regulatory protein Bap60. The *Bap60* gene is expressed from both maternal and zygotic expression transcripts and increases throughout development (Möller *et al*, 2005), corresponding with the protein abundance profiles observed here. Together, these data highlight the quantitative resolution of single-embryo analyses with SP3, as well as demonstrating the potential for new applications in screening of closely related individuals, such as isogenic libraries where expression differences are expected to be small.

The findings of this work have important implications toward advancing the development of proteomics as a supplementary tool for understanding complex biological systems, in particular for integrative studies combining proteome data with that from multiple streams, such as genomics, transcriptomics, and metabolomics. Of particular note is the increased sensitivity provided by SP3 that reduces the absolute sample requirements for proteomics, permitting splitting of material to acquire sample-matched data. As MS instrumentation continues to advance and gains in sensitivity are made, we expect that integration of SP3 with these cutting-edge platforms will push proteomics toward achieving complete proteome profiling from ultrasensitive samples. In the future, we envision extension of these analyses to screening of individuals within genetic libraries, clinical fluid or tissue samples (before or after paraffin embedding), or previously inaccessible ultrasensitive applications. These extensions will be further aided by the cost efficiency of SP3 (< 1 cent per assay based on consumable cost), extending its global reach in research studies. The compatibility and ease of adaption of SP3 protocols to robotics or microfluidic platforms will decrease these costs further, while increasing throughput and automation. This collection of benefits positions SP3 as an incredibly robust single-tube proteomics method integrating all steps from cell lysis to peptide fractionation.

## Materials and Methods

### Paramagnetic beads

In all experiments with SP3, we utilize commercially available beads that carry a carboxylate moiety. We have tested and verified the

protocols used in this study with beads from Beckman Coulter (Ampure XP, CAT# A63880), CleanNA (Clean PCR, CAT# CPCR1300), ReSyn Biosciences (MagReSyn, CAT# MR-CBX005), and Thermo Fisher (Sera-Mag Speed Beads, CAT# 09-981-121, 09-981-123) with a variety of reagents (Fig 1B; Supplementary Fig S1A). In all cases, beads have an average diameter of 1 μm and are coated with a hydrophilic surface. For all SP3 experiments in this manuscript, a 1:1 combination mix of the two types of Sera-Mag speed beads is used. Beads are rinsed with water prior to use and stored at 4°C at a stock concentration of 10 μg/μl. Magnetic racks used in all experiments were designed and manufactured in-house (Supplementary Fig S1B and C). Neodymium magnets used in racks were purchased from Supermagnete (Germany).

### Handling of *Drosophila* embryo samples

Staged embryos from line RAL-859 (Bloomington #25210) were collected on agar plates at 25°C following standard protocols. In brief, following three 1-h pre-lays (embryo collections that serve to synchronize the developmental stages of the final embryo collection), embryos were collected during 2-h windows and aged at 25°C to the appropriate stage [2–4 h, 2–2.5 h, 3.5–4 h, or 10–12 h after egg lay (AEL)]. The embryos were then rinsed with water, dechorionated by incubating in 3% NaOCl (50% bleach) for 2 min, washed with PBT (PBS with 0.1% Triton), and air-dried at room temperature. Single embryos were then transferred to individual PCR tubes and flash frozen in liquid nitrogen. Details of yeast, HeLa, mESC, and NP culture are given in the Supplementary Information.

### Cell lysis, protein reduction, and alkylation

In all experiments involving the use of yeast, lysis was performed using a mechanical bead beating procedure in 1% SDS-containing buffer. In all HeLa, mESC, and NP experiments, lysis was performed using 1% SDS-containing buffer with Benzonase treatment to shear chromatin. Details of yeast, HeLa, mESC, and NP lysis, reduction, and alkylation protocols are given in the Supplementary Information.

*Drosophila* embryos were lysed using a solution-based procedure with sonication. Single embryos in PCR tubes were re-suspended in 20 μl of lysis buffer containing 20 μg of SP3 beads. Lysis buffer was composed of 1% SDS (Bio-Rad), 1× cOmplete Protease Inhibitor Cocktail-EDTA (Roche), 5 mM EDTA, 5 mM EGTA, 10 mM NaOH, and 10 mM DTT, in 10 mM HEPES buffer at pH 8.5 (Sigma). The lysis solution was combined with 20 μl of neat trifluoroethanol (Sigma) and sonicated for 15 min in a Bioruptor (Diagenode) for 10 cycles (30 s on, 30 s off) on the setting 'high'. The Bioruptor was operated in the absence of active cooling to allow the water bath to heat and facilitate lysis and shearing of chromatin. To neutralize the lysis solution, 0.75 μl of 0.1% formic acid was added. Samples were then heated at 95°C for 5 min and placed on ice before proceeding with reduction and alkylation steps. (Details of reduction and alkylation can be found in the Supplementary Information.)

### Protein SP3 protocol

Details of SP3 optimization experiments for protein binding as well as digestion conditions are given in the Supplementary Information.

Step-by-step protocols for performing SP3 can also be found in the Supplementary Information.

The Sera-Mag SP3 bead mix that had been prepared as discussed above was stored as a stock at a concentration of 10 μg/μl. Unless otherwise noted, all reactions are carried out in untreated PCR tubes (Ratiolab). To each protein mixture to be treated, 2 μl of this bead stock (20 μg) was added and pipette mixed to generate a homogeneous solution. Although 20 μg of beads exceeds the required amount given the binding capacity of 1 μg of beads per 100 μg of protein determined above, it is beneficial to retain the absolute bead concentration at values > 0.1 μg/μl in solution to promote bead aggregation as the reaction progresses (e.g., 20 μg of beads after addition of 195 μl in peptide SP3 gives 0.1 μg/μl in the total volume of 200 μl). The bead–protein mixture was then acidified (pH ~2) through the addition of formic acid, and acetonitrile (100% stock) was added to a reach a final concentration of 50% (v/v) of the total volume. Mixtures were incubated upright for 8 min at room temperature and then placed on a magnetic rack for a further 2 min. While on the magnet, the supernatant was removed and discarded. The beads were rinsed through addition of 200 μl of 70% absolute ethanol, incubated for 30 s, and the supernatant discarded. This step was repeated one further time. Beads were then rinsed one further time with 180 μl of 100% acetonitrile, incubated for 30 s, and the supernatant discarded. All rinses were carried out while tubes were mounted on the magnetic rack. Rinsed beads were then reconstituted in aqueous buffer (e.g., 5 μl of 50 mM HEPES pH 8), pipette mixed, and incubated for 5 min at room temperature to elute proteins. The composition of the elution buffer can be catered to the desired downstream protocol.

### Peptide SP3 protocol

Details of SP3 optimization experiments for peptide binding as well as fractionation conditions are given in the Supplementary Information. Step-by-step protocols for performing SP3 can also be found in the Supplementary Information.

When peptide mixtures did not originate from a previous SP3 digest and thus did not contain beads, 2 μl of beads from a 10 μg/μl stock (20 μg) was added and pipette mixed. When peptide mixtures were derived from an SP3 digest, bead–peptide solutions were pipette mixed to re-suspend the beads that had settled during the digestion procedure. Unless otherwise noted, all reactions are carried out in untreated PCR tubes. To each tube, 100% acetonitrile was added to achieve a final concentration > 95% (e.g., 5 μl of bead-protein mixture + 195 μl of 100% acetonitrile). Mixtures were incubated for 8 min at room temperature and following this placed on a magnetic rack for a further 2 min. The supernatant was discarded, and the beads rinsed one time with 180 μl of 100% acetonitrile. Rinsed beads were reconstituted in an aqueous solution (typically H₂O) containing 2% dimethylsulfoxide (DMSO). The volume used for elution can be catered for the downstream application and the expected concentration of the peptide mixture (e.g., 10 μl). Mixtures were pipette mixed and incubated for 5 min at room temperature. Tubes were placed on a magnetic rack and eluted peptides recovered. Prior to analysis with MS and after removal from the paramagnetic beads, peptides solutions were acidified with formic acid.

### Mass spectrometry data acquisition

Experiments involving the analysis of limited amounts of material (HeLa and *Drosophila*) were carried out on an Orbitrap Velos Pro MS system (Thermo Scientific) equipped with a nanoAcquity liquid chromatography system (Waters). Injected peptides were trapped on Symmetry C18 columns (180 μm x 20 mm). After trapping, gradient elution of peptides was performed on a C18 (nanoAcquity BEH130 C18, 75 μm x 200 mm, 1.7 μm) column. For single-shot samples where extended analysis was used, elution was performed with a gradient of mobile phase A (99.9% water and 0.1% formic acid) to 25% B (99.9% acetonitrile and 0.1% formic acid) over 190 min, and to 40% B over 40 min, for a final length of 265 min. For samples fractionated with high-pH reversed-phase, 145-min gradient runs were used (Supplementary Methods). For SP3 fractionated samples, 180-min gradients were utilized where the percentage of B was ramped to 25% over 120 min and to 40% B over 40 min, for a final length of 210 min.

Data acquisition on the Orbitrap Velos Pro MS was carried out using a data-dependent method. The top 15 precursors were selected for MS2 analysis after collisional induced fragmentation (CID). Survey scans covering the mass range of 350–1,500 were acquired at a resolution of 30,000 (at $m/z$ 400) with a maximum fill time of 500 ms and an AGC target value of 1e6. MS2 scans were acquired with a maximum fill time of 50 ms and an AGC target value of 1e4 with an isolation window of 2.0 $m/z$. CID fragmentation was induced with an NCE of 40, an activation time of 10 ms, and an activation Q of 0.250. Dynamic exclusion was set to exclude previously selected precursors for a total of 60–90 s depending on gradient length. Charge state exclusion was set to ignore unassigned, 1, and 4 and greater charges. MS1 data were acquired in profile mode, whereas MS2 data were obtained in centroid format. Further details of conditions used for MS analysis are given in the Supplementary Information.

### Data analysis

Details of data processing, statistical validation and testing, and biological feature extraction are given in the Supplementary Information.

### Data availability

All raw data and protein and peptide identification tables associated with this manuscript can be downloaded from Chorus (https://chorusproject.org) under the title "Enhanced workflows with paramagnetic beads for ultrasensitive proteomics". All scripts associated with this manuscript are available upon request.

**Supplementary information** for this article is available online: http://msb.embopress.org

### Acknowledgements
The authors would like to acknowledge valuable contributions, input, and discussions from Raeka Aiyar, Kristina Dzeyk, Stefan Leicht, Marco Hennrich, Lida Radan, and Joanna Kirkpatrick. The authors would also like to acknowledge Lucia Ciglar and Raquel Ferreres for valuable help during embryo collections.

## Author contributions

CH conceived the idea, designed and carried out the experiments and data analysis. SF carried out experiments. DG aided in experimental design and carried out experiments. CH and JK wrote the manuscript. EF and LS aided in experimental design and contributed to writing of the manuscript. CH is funded by a EU Marie Curie Actions Cofund grant for EIPOD fellows awarded to EMBL.

## Conflict of interest

The authors declare that they have no conflict of interest.

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
