## [Review Process File · Molecular Systems Biology]

Ultrasensitive proteome analysis using paramagnetic bead technology

Christopher Hughes, Sophia Foehr, David Garfield, Eileen E.M. Furlong, Lars M. Steinmetz, Jeroen Krijgsveld

Corresponding author: Jeroen Krijgsveld, European Molecular Biology Laboratory

Review timeline:

Submission date:	27 July 2014
Editorial Decision:	27 August 2014
Revision received:	09 September 2014
Accepted:	07 October 2014

Editor: Maria Polychronidou

Transaction Report:

1st Editorial Decision

27 August 2014

Thank you again for submitting your work to Molecular Systems Biology. We have now heard back from the three referees who agreed to evaluate your manuscript. As you will see from the reports below, the referees acknowledge that the presented methodology is a potentially valuable contribution to the field. However, they raise a series of concerns, which should be carefully addressed in a revision of the manuscript.

Without repeating all the comments listed below, some of the more fundamental issues are the following:

- Reviewer #1 points out that additional analyses on the proteomic changes during *Drosophila* embryo development will significantly enhance the impact of the biological findings.
- The advantages and improved performance of the presented approach compared to existing methods need to be more conclusively demonstrated.
- The reviewers refer to the need to describe the protocol and related experimental conditions in better detail.

If you feel you can satisfactorily deal with these points and those listed by the referees, you may wish to submit a revised version of your manuscript. Please attach a covering letter giving details of the way in which you have handled each of the points raised by the referees. A revised manuscript will be once again subject to review and you probably understand that we can give you no guarantee at this stage that the eventual outcome will be favorable.

Reviewer #1:

The authors report a newly developed method, Single-Pot Solid-Phase-enhanced Sample Preparation (SP3) that is applicable in ultrasensitive proteomic analysis and is also compatible with chemical labelling strategies.

The results that they present demonstrate nicely several advantages of employing SP3 for proteomic studies, including high recovery, high reproducibility and high flexibility. The method is also easy to handle and is less sensitive to the detergents and salts that are required during sample preparation. The newly developed method was then applied for the characterization of 1000 HeLa cells and single *Drosophila* embryos. Such a sensitive methodology will definitely be of great interest to specialists, and could benefit the whole proteomic field. It is of particular topical interest, as other very recently published studies also focus upon minute sample preparation techniques. However, I have some difficulties with the overall presentation of the manuscript, as it combines a novel method for enhanced and fast sample preparation with a proteomic dataset. Regrettably, the latter does not entirely meet the standards of molecular systems biology, as it lacks any real system-level data and analysis. More data supporting the time-dependent proteomic changes during embryo development, and also careful biological validation, are required. Moreover, in its current form, the truly important "methods" part of the manuscript is not entirely satisfactorily illustrated (see also below).

Besides these principal concerns I would like to point out several other critical points that the authors should consider:

Major points:

1. The authors compared the protein or peptide recovery of different sample preparations on the basis of the image density on SDS-polyacrylamide gels stained with Coomassie blue and of the base peak intensity found in LC-MS/MS analysis. Neither of these approaches is sensitive to the loss of proteins or peptides with medium to low abundance. Although Figures E6 and E7 have clearly shown comparable data qualities, and similar chemical/physical properties, of peptides obtained by using SP3 and FASP, it is still difficult to explain the significant decrease in base peak intensity of samples prepared by FASP and precipitation compared with the one prepared by SP3 (Figure E1c). (This latter point may prove controversial.) Was it because of general peptide loss, or were there peptides with specific chemical/physical properties to be recovered exclusively by SP3? What kinds of mechanisms are involved in the interaction between proteins/peptides and magnetic beads? Is this approach applicable for all kinds of HILIC-based beads? It may be necessary to examine these issues more carefully.
2. The authors have included a variety of conditions for optimizing the SP3 method, both in the main text and in the Expanded Methods. Nevertheless, it remains confusing and difficult for readers to trace which exact conditions were used in individual experiments. The graphical workflow shown in Figure 1a contains too little information and should be re-drawn, including all conditions used. In addition, the optimized condition for protein SP3 is 50% acetonitrile under acidic conditions (pH ~2). The authors should specify which acid was used, and in what concentration.
3. In this context it is somewhat disappointing that the authors provide a very poor methodological description of the cell lysis in the main text. In my opinion this might be a strength of this particular manuscript and the method behind it, if the sample-preparation technique is compatible with a variety of detergents including SDS. The latter, however, is applied at least to the *Drosophila* samples - but can it be applied to yeast and HeLa cells as well?
4. To make a greater scientific impact in the field of systems biology, much more discussion and validation should be included in the manuscript instead of the simple comparison of protein expression correlation, among developmental stages, that the authors provide. It is also unclear what rationale lay behind the choice of proteins with >1-fold (Fig 6a) or >0.75-fold (Fig 6b) expression difference for comparison.

Minor points

5. It would be interesting to have a direct and comprehensive (multi-parameter) comparison of SP3 with FASP or other "conventional" methods for preparing low-amount samples such as those from 1000 HeLa cells. This could confirm whether SP3 indeed outperforms any existing methods for low-amount samples.
6. The authors should explain why the protein abundance (log(iBAQ)) profile of 1000 cells is left-

shifted compared with other samples (Fig E8).

7. Based on the manufacturer's data sheet, the magnetic beads employed in this study are stable only at pH 4-11. It would be valuable if the authors could show the results supporting their assertion that preparation under pH ~2 is optimized for protein SP3 protocol.

8. Page 11, line 14, "Figure E8b": This should read "Figure E8a".

9. Page 11, line 16, "Figure E8a": This should read "Figure E8b".

10. The Venn diagram shown in Figure 4b and E14a is confusing for the reader. The colour coding makes it difficult to follow which component belongs to which condition. The authors should stick to the classical representation for Venn diagrams.

11. The manuscript provides a wealth of figures that show the nice results that the authors have obtained by their method. Unfortunately, the number of supplementary figures makes the manuscript confusing to read. This should be solved by reducing the number of figures such as to focus on the most important ones.

Reviewer #2:

The manuscript by Hughes and colleagues presents a new method which facilitates proteome analysis. The idea is termed Single-Pot Solid-Phase-enhanced Sample Preparation (SP3). Paramagnetic beads with carboxylate coating act by a HILIC-like (or ERLIC-like) mechanism to immobilize proteins from lysates. After immobilization, protein can be reduced and alkylated, and then proteolyzed. Even further processing (such as alkylation with methylating reagents or isobaric tags) are possible. The authors demonstrate fairly exhaustively the ability of the SP3 method to work reproducibly with samples of normal quantities as well as in sample-limited applications. Single cell embryos analyses from *Drosophila* are presented. Overall, I found this to be a compelling paper with strong potential to change the way we process samples for proteomics applications.

Specific comments:

1) This reviewer is interested in the precise amount of time it takes to perform the SP3 method. For example, FASP seems to be a very rapid workflow and occurs in a single tube, but the spins can sometimes take hours (depending on the membrane), and unless one is working with more than 24 samples, it can be more work to use the FASP protocol. Could the authors comment on the time it would take to process 12 samples in parallel? Digestion is obviously overnight. This would seem to be an advantage of the SP3 method.

2) Figure 1C describes a wealth of conditions where the SP3 method can function. The data behind this table are in Supplementary Figures. Given that this is as important as the ability to work with very small sample amounts, it would be a good idea to remove one of the later figures and create a main-text figure that shows some of the primary data. It is of great interest to most readers.

3) What does "greater than 1-fold" difference mean in Fig. 6? Do the authors mean 2-fold difference of Log₂ ratio (which is a value of 1)? A 1-fold difference is no change. Finally, Fig. 6 gets the message across, but it seems like a dendrogram of the clustered samples may help to highlight the similarities and differences in these data.

Reviewer #3:

This manuscript describes a novel sample processing workflow tailored to handle limited sample sizes (<1µg) with minimal sample losses. All sample processing steps from cell lysis to peptide elution for MS analysis are performed in a single tube thereby minimizing sample losses. Importantly, this workflow is compatible with a wide range of additives to improve sample solubility, cell lysis and/or digestion including strong detergents such as SDS, Triton X and NP-40. Proteins or peptides are first immobilized at the surface of magnetic beads covered by a hydrophilic coating by varying the concentration of organic buffers (e.g. ACN). This immobilization step facilitates downstream sample processing for subsequent elution in aqueous buffers. The application of this new protocol is demonstrated using small amount of cell extracts, whereby more than 1250 and 2900 proteins were identified using only 1000 HeLa cells and single *Drosophila* embryos,

respectively. The originality of this work stems from a new and simple sample processing protocol enabling the cleanup of protein/peptide extracts containing detergent, and the impressive analytical performances obtained from sub microgram amounts of samples. Overall, these studies are well executed and provide valuable information on efficient sample processing for a wide range of proteomics applications. I recommend publication pending minor revision as outlined below:

1. In the abstract, introduction and result sections, the authors indicated that the SP3 protocol facilitates ultrasensitive proteomics analysis by outperforming existing protocols in terms of efficiency, scalability, speed, throughput, and flexibility. While comparisons are made with existing protocols (e.g. FASP), it is not clear to this reviewer how SP3 compares with FASP for limited amount of material, and if the claim of outperforming other protocols is fully supported by appropriate data. For example, Fig E6a shows comparable results between FASP and SP3 for duplicate aliquots of a yeast whole-cell lysate (10 g), but no data are provided for smaller amounts of material. It would be pertinent to make the comparison of identification between SP3 and FASP or StageTip (Kulak et al, 2014) for HeLa digests ranging from 500,000 to 1,000 cells.
2. The description of the sample processing protocol is not entirely clear, and different parts of the workflow are scattered across multiple sections often applied to different samples. I was not able to find a clear step by step procedure on how to process sample with SP3 from cell lysis to MS analysis. A more straightforward description of the optimized SP3 protocol is required in the extended materials and methods section for users interested in applying this protocol for proteomics samples.
3. Very often MS results are represented as overlaid XIC's. This comparison is visually attractive but not very informative. Can the authors provide label-free quantitation as scatter plots showing peptide intensities from different samples? These analyses can be performed using MaxQuant.
4. Can the authors provide data on sample binding capacity (protein and peptide) on magnetic beads? Different peptide/protein loadings should be provided to clearly illustrate what is the maximum binding capacity (preferable in the presence of different detergents to illustrate their effect). I'm surprised to see that the HILIC media has a binding capacity of 100 ug of protein per 1 ug of beads knowing that C18 media typically has a binding capacity of approximately 5% on a mass basis.

1st Revision - authors' response

09 September 2014

Reviewer #1:

The authors report a newly developed method, Single-Pot Solid-Phase-enhanced Sample Preparation (SP3) that is applicable in ultrasensitive proteomic analysis and is also compatible with chemical labelling strategies.

The results that they present demonstrate nicely several advantages of employing SP3 for proteomic studies, including high recovery, high reproducibility and high flexibility. The method is also easy to handle and is less sensitive to the detergents and salts that are required during sample preparation. The newly developed method was then applied for the characterization of 1000 HeLa cells and single *Drosophila* embryos. Such a sensitive methodology will definitely be of great interest to specialists, and could benefit the whole proteomic field. It is of particular topical interest, as other very recently published studies also focus upon minute sample preparation techniques.

However, I have some difficulties with the overall presentation of the manuscript, as it combines a novel method for enhanced and fast sample preparation with a proteomic dataset. Regrettably, the latter does not entirely meet the standards of molecular systems biology, as it lacks any real system-level data and analysis. More data supporting the time-dependent proteomic changes during embryo development, and also careful biological validation, are required. Moreover, in its current form, the truly important "methods" part of the manuscript is not entirely satisfactorily illustrated (see also below).

Besides these principal concerns I would like to point out several other critical points that the authors should consider:

Major points:

1. The authors compared the protein or peptide recovery of different sample preparations on the basis of the image density on SDS-polyacrylamide gels stained with Coomassie blue and of the base peak intensity found in LC-MS/MS analysis. Neither of these approaches is sensitive to the loss of proteins or peptides with medium to low abundance. Although Figures E6 and E7 have clearly shown comparable data qualities, and similar chemical/physical properties, of peptides obtained by using SP3 and FASP, it is still difficult to explain the significant decrease in base peak intensity of samples prepared by FASP and precipitation compared with the one prepared by SP3 (Figure E1c). (This latter point may prove controversial.) Was it because of general peptide loss, or were there peptides with specific chemical/physical properties to be recovered exclusively by SP3? What kinds of mechanisms are involved in the interaction between proteins/peptides and magnetic beads? Is this approach applicable for all kinds of HILIC-based beads? It may be necessary to examine these issues more carefully.

Author response: We appreciate the questions raised by the reviewer and agree with the comments based on the limited ability of SDS-PAGE and single-shot chromatograms to illustrate trends on a global proteomic scale. However, as also noted by the reviewer, the in-depth comparative information provides sufficient resolution to assert our claims. The differences in the recovery between the FASP, precipitation, and SP3 protocols are non-specific losses. If the losses were due to a specific loss/enrichment of a subset of peptides or proteins in FASP or SP3, this would be detectable in the in-depth analyses presented here. In all of the mentioned experiments, including the in-depth comparison, we are still dealing with a relatively small amount of starting material (~10 ug). The efficiency of precipitation is known to drop as the amount of material is reduced. FASP has also been demonstrated to suffer from losses due to non-specific adsorption to the filter unit, as well as reductions in efficiency when material is reduced. Some studies (from the lab that developed FASP) have even estimated the recovery of material in the FASP protocol to be as low as 50% of starting (Wisniewski, J. Analytical Biochemistry). In the case of FASP, this will not necessarily lead to a reduction in the number of identifications you can obtain in an analysis provided your amount of starting material is sufficient enough to overcome these losses. This is demonstrated by the in-depth data where we are still able to obtain comparable depth of coverage to SP3 in spite of these losses. We have added these points and referenced them in the main text (Pg. 8 Para. 2, Pg. 11 Para. 1, Pg. 12 Para. 3).

The mechanism of interaction of proteins and peptides with the beads is the same as those found in HILIC or ERLIC. Proteins or peptides are trapped in a solvation layer on the hydrophilic surface of the bead driven by the addition of organic solvent (acetonitrile). Modulation of the pH can change the character of this interaction, where at high-pH values the negatively charged carboxylate surface of the bead provides a repulsive interaction with proteins or peptides. The charged nature of the proteins or peptides increases their polarity and enhances their transition into the hydrophilic solvation layer on the bead surface. When the concentration of organic is increased, this interaction is sufficient to overcome any repulsion from the negatively charged bead surface. We expect that SP3 would indeed work with conventional HILIC or ERLIC phases. Towards this goal we have attempted experiments using commercial versions of paramagnetic weak- and strong-anion exchange beads. However, we have been unable to source a commercially available bead that carries a compatible combination of surface and functional group chemistries, and thus have been unable to fully test these possibilities. To clarify these details for the reader, we have expanded the information in the text (Pg. 7 Para. 1).

2. The authors have included a variety of conditions for optimizing the SP3 method, both in the main text and in the Expanded Methods. Nevertheless, it remains confusing and difficult for readers to trace which exact conditions were used in individual experiments. The graphical workflow shown in Figure 1a contains too little information and should be re-drawn, including all conditions used. In addition, the optimized condition for protein SP3 is 50% acetonitrile under acidic conditions (pH ~2). The authors should specify which acid was used, and in what concentration.

Author response: In lieu of re-drawing Figure 1a, we have elaborated on the main steps in the figure legend and main text of the results (Pg. 7 Para. 1, 2, Pg. 8 Para. 1). In feedback from previous reviewers and conference attendees to whom this work has been presented, an expanded visual representation of the SP3 protocol becomes confusing for the reader due to the large number of elements. In an effort to improve the clarity of the methods and protocol in the absence of an expanded figure, we have significantly edited the main and Expanded view text. In addition, we have now provided step-by-step details for performing SP3 and support methods (e.g. cell lysis) in a variety of situations as part of the Expanded protocols section.

3. In this context it is somewhat disappointing that the authors provide a very poor methodological description of the cell lysis in the main text. In my opinion this might be a strength of this particular manuscript and the method behind it, if the sample-preparation technique is compatible with a variety of detergents including SDS. The latter, however, is applied at least to the *Drosophila* samples - but can it be applied to yeast and HeLa cells as well?

*Author response: Although we feel we have given extensive detail of the *Drosophila* lysis as part of the methods in the main text, we agree that the lysis methods deserve more attention. To remedy this, we have expanded the detail in the Expanded view methods for cell lysis, as well as adding these methods in step-by-step format as part of the Expanded protocols section. This is indeed an important detail of the manuscript, as detergent-based (SDS) lysis conditions are used in every experiment presented. In addition, the *Drosophila* lysis protocol has the added benefit of eliminating the need for mechanical disruption, not just for embryos, but also for yeast cells, which are notoriously difficult to lyse. To highlight these points, we have included more discussion of the chosen lysis method, as well as how the flexibility of SP3 benefits us in these situations (Pg. 13 Para 1, Pg. 15 Para 1, Pg. 19 Para. 2, Pg. 20 Para. 1,2).*

4. To make a greater scientific impact in the field of systems biology, much more discussion and validation should be included in the manuscript instead of the simple comparison of protein expression correlation, among developmental stages, that the authors provide. It is also unclear what rationale lay behind the choice of proteins with >1-fold (Fig 6a) or >0.75-fold (Fig 6b) expression difference for comparison.

Author response: We appreciate the concerns of the reviewer, but we feel this is an oversimplification of the results we have provided. Nevertheless, we have provided further discussion of the results including validation information sourced from public repositories and literature to strengthen the manuscript (Pg. 25-29, Pg. 16 Para. 2, Pg. 18 Para. 2, Pg. 20 Para. 1, Figure E13 d,e).

*In support of the data, the presented work provides a first look into the expressed proteomes during developmental specific stages in *Drosophila*. Currently there are few studies that catalog the *Drosophila* proteome during development, and none that do so to the depth presented here. Perhaps most importantly we have extended this to making a quantitative comparison of protein levels during *Drosophila* development at a single-embryo resolution, generating an entirely unique data set that is unmatched in the field. Although antibody-based methods offer the ability to screen at a single embryo resolution and give the added benefit of localization information, these approaches are limited in the number of candidates they can assay. With the MS approach presented here we successfully track >2,900 proteins across multiple individuals and conditions. This permitted the simultaneous comparison of stage-specific dynamics, as well as inter-individual patterns. These data revealed diverse changes in stage-specific proteomes in pathways related to transcription and translation regulation of developmental processes such as neurogenesis, as well as the maternal-to-zygotic transition. At multiple points in the manuscript we have validated these data using gene expression patterns from a variety of technologies (RNAseq, RNA in-situ hybridization). Although these data come from public sources, this does not detract from the quality of the results validated by them. Perhaps the most attractive quality of working in *Drosophila* is the wealth of cataloged, standardized, and high-quality data available in multiple public resources. Re-acquisition of these data would be redundant.*

While we agree with the reviewer that the figure in question (Figure 6) is indeed a simple protein correlation, this method serves the purpose that we are trying to achieve. The goal of the figure is to demonstrate the ability to map individual embryos based on their protein expression profile. This

figure highlights both the quantitative accuracy and resolution of the SP3 method, and also the ability to detect expression differences when total proteome variation is small. We have added additional validation information in support of the differences we see between the stages based on RNAseq data taken from sectioned embryos covering the 2-4 hour window (Figure E13 d,e, Pg. 19 Para. 2). In the case of the original data presented, the thresholds were chosen based on the fold-change difference cut-off that retained a sufficient data quantity to generate a robust correlation. We have clarified this information in the main text (Pg. 18 Para. 2, Pg. 19 Para. 1).

Together, the combination of in-depth protein information with coupled gene expression data at a single-embryo resolution and mapping to developmentally relevant processes (maternal-to-zygotic transition) provides a unique, systems-level look into *Drosophila* development.

Minor points

5. It would be interesting to have a direct and comprehensive (multi-parameter) comparison of SP3 with FASP or other "conventional" methods for preparing low-amount samples such as those from 1000 HeLa cells. This could confirm whether SP3 indeed outperforms any existing methods for low-amount samples.

Author response: We appreciate and agree with the reviewers concerns in this area. In a previous version of this manuscript we had completed a comparative analysis of SP3 and FASP for a starting material of 1,000 HeLa cells that is included below for your consideration. It clearly demonstrates the superiority of SP3 in this instance compared with FASP. However, we hesitate to include these data in the main manuscript, as we believe it will be met with significant criticism from the proteomics community, and has been in the past by reviewers. These criticisms largely stem from the fact that FASP is known to have poor performance for low sample amounts, and is thus rarely applied in these cases.

There is currently no 'gold-standard' protocol for doing ultra-sensitive proteomics. Of the protocols that have been presented, none have seen widespread adoption by the community, due to either limitations in sensitivity or the requirement for specialized equipment. Recently, a high-efficiency approach based on the StageTip protocol was introduced, termed iST, for doing proteomics on low input amounts (Kulak, N. et al. Nature Methods). The iST method was developed by the lab that also presented FASP, promoting it as a superior solution for high-efficiency proteomics processing. Given the familiarity of the StageTip protocol and the high-impact level of the journal in which it is published, we feel that this method presents the most appropriate comparison that will be relevant to a broad readership. In the published iST manuscript, an ultra-sensitive screen of differing numbers of HeLa cells was performed (Supplementary Figure 1d). Given that these data were acquired in the lab that developed iST, using the optimal method and instrument settings, re-acquisition of the data by us would be repetitive. We have elaborated on this comparison in the current version of the manuscript (Pg. 12 Para. 2, Pg.13 Para. 1). Unfortunately, Kulak et al. did not release the raw data, or processed values for this specific part of the manuscript, limiting the depth of the comparison we can provide. However, we feel that based on the values presented we have successfully demonstrated the superiority of SP3. We have further elaborated in the discussion about these aspects as well (Pg. 20 Para. 2,3, Pg. 22 Para. 2).

6. The authors should explain why the protein abundance (log(iBAQ)) profile of 1000 cells is left-shifted compared with other samples (Fig E8).

Author response: The 1,000 cell sample is shifted due to the nature of the iBAQ calculation. The sampling of the proteome in the 50,000 and 5,000 cell samples is relatively equivalent, based on sequence coverage and peptide intensities. As a result, the numbers of peptides and intensities being used in the iBAQ calculation is more-or-less equivalent. In the 1,000 cell sample, this depth of sampling is much lower, both at the coverage and intensity level. Thus, the summed intensity of observed peptides drops, while the theoretical number of observable peptides remains the same, thereby shifting the distribution to lower values. This is a limitation of the iBAQ approach when an insufficient portion of the proteome has been sampled. We have noted this in the main text (Pg. 12 Para. 1).

7. Based on the manufacturer's data sheet, the magnetic beads employed in this study are stable only at pH 4-11. It would be valuable if the authors could show the results supporting their assertion that preparation under pH ~2 is optimized for protein SP3 protocol.

Author response: We appreciate the concerns of the reviewer, however we have opted not to include all of these data in a Figure format. Due to the number of conditions tested, this would require adding a significant amount of additional figure and discussion space, while giving minimal additional value to the reader. The most relevant of the mentioned data is already present in the manuscript (Figure E2b for proteins, Figure E4a-b for peptides). These data demonstrate the differences between enrichments at different acetonitrile concentrations, as well as when acidification is used. Although the beads are listed as stable at only pH 4-11, these are guidelines directed at extended storage or use. In the SP3 protocol, the beads are exposed to acidic conditions for a total of 8 minutes, during which we have not noticed any bead degradation or loss of fidelity in the protocol.

8. Page 11, line 14, "Figure E8b": This should read "Figure E8a".

Author response: Changed in manuscript.

9. Page 11, line 16, "Figure E8a": This should read „ Figure E8b".

Author response: Changed in manuscript.

10. The Venn diagram shown in Figure 4b and E14a is confusing for the reader. The colour coding makes it difficult to follow which component belongs to which condition. The authors should stick to the classical representation for Venn diagrams.

Author response: We agree and have re-drawn the corresponding figure elements.

11. The manuscript provides a wealth of figures that show the nice results that the authors have obtained by their method. Unfortunately, the number of supplementary figures makes the manuscript confusing to read. This should be solved by reducing the number of figures such as to focus on the most important ones.

Author response: We appreciate the reviewers comment and we have attempted to condense the supplementary figures where possible. However, given the broad subject matter of the manuscript, the presented data are of interest to the potential readership. The data on optimization, protein and peptide distributions, and mass spectrometry properties will be of value to proteomics experts, as well as those looking to utilize SP3 in their own lab. The data on the Drosophila embryo screens is essential for providing complete transparency in relation to the conclusions we are making.

Reviewer #2:

The manuscript by Hughes and colleagues presents a new method which facilitates proteome analysis. The idea is termed Single-Pot Solid-Phase-enhanced Sample Preparation (SP3). Paramagnetic beads with carboxylate coating act by a HILIC-like (or ERLIC-like) mechanism to

immobilize proteins from lysates. After immobilization, protein can be reduced and alkylated, and then proteolyzed. Even further processing (such as alkylation with methylating reagents or isobaric tags) are possible. The authors demonstrate fairly exhaustively the ability of the SP3 method to work reproducibly with samples of normal quantities as well as in sample-limited applications. Single cell embryos analyses from *Drosophila* are presented. Overall, I found this to be a compelling paper with strong potential to change the way we process samples for proteomics applications.

Specific comments:

1) This reviewer is interested in the precise amount of time it takes to perform the SP3 method. For example, FASP seems to be a very rapid workflow and occurs in a single tube, but the spins can sometimes take hours (depending on the membrane), and unless one is working with more than 24 samples, it can be more work to use the FASP protocol. Could the authors comment on the time it would take to process 12 samples in parallel? Digestion is obviously overnight. This would seem to be an advantage of the SP3 method.

Author response: We have attempted to highlight the processing time requirements for SP3 in the main text (Pg. 8, Para. 1), as well as promoting the potential extensions to high throughput screens in a 96-well format (Pg. 22 Para. 2) without requiring specialized equipment. To answer the question of the reviewer, processing samples in parallel adds no extra time to the protocol compared to treating a single sample. For example, we regularly process batches of 96 samples in parallel in a well-plate format, with SP3 (protein and peptide each) requiring just 15 minutes to complete.

2) Figure 1C describes a wealth of conditions where the SP3 method can function. The data behind this table are in Supplementary Figures. Given that this is as important as the ability to work with very small sample amounts, it would be a good idea to remove one of the later figures and create a main-text figure that shows some of the primary data. It is of great interest to most readers.

*Author response: We agree with the comments of the reviewer regarding the importance of the diverse nature of conditions compatible with SP3, as this is in fact one of the most attractive features of the protocol. However, we have opted to keep the relevant data (with the exception of Figure 1b) in the enhanced view to avoid drawing too much attention from the *Drosophila* screen. Instead, we have attempted to highlight these data at multiple points in the text (Pg. 7 Para. 2, Pg. 9 Para. 1, Pg. 19 Para. 2, Pg. 20 Para. 1,2).*

3) What does "greater than 1-fold" difference mean in Fig. 6? Do the authors mean 2-fold difference of Log₂ ratio (which is a value of 1)? A 1-fold difference is no change. Finally, Fig. 6 gets the message across, but it seems like a dendrogram of the clustered samples may help to highlight the similarities and differences in these data.

Author response: This is poor wording, and we thank the reviewer for bringing this to our attention. This has been amended in the main text.

Reviewer #3:

This manuscript describes a novel sample processing workflow tailored to handle limited sample sizes (<1µg) with minimal sample losses. All sample processing steps from cell lysis to peptide elution for MS analysis are performed in a single tube thereby minimizing sample losses. Importantly, this workflow is compatible with a wide range of additives to improve sample solubility, cell lysis and/or digestion including strong detergents such as SDS, Triton X and NP-40. Proteins or peptides are first immobilized at the surface of magnetic beads covered by a hydrophilic coating by varying the concentration of organic buffers (e.g. ACN). This immobilization step facilitates downstream sample processing for subsequent elution in aqueous buffers. The application of this new protocol is demonstrated using small amount of cell extracts, whereby more than 1250 and 2900 proteins were identified using only 1000 HeLa cells and single *Drosophila* embryos, respectively. The originality of this work stems from a new and simple sample processing protocol enabling the cleanup of

protein/peptide extracts containing detergent, and the impressive analytical performances obtained from sub microgram amounts of samples. Overall, these studies are well executed and provide valuable information on efficient sample processing for a wide range of proteomics applications. I recommend publication pending minor revision as outlined below:

1. In the abstract, introduction and result sections, the authors indicated that the SP3 protocol facilitates ultrasensitive proteomics analysis by outperforming existing protocols in terms of efficiency, scalability, speed, throughput, and flexibility. While comparisons are made with existing protocols (e.g. FASP), it is not clear to this reviewer how SP3 compares with FASP for limited amount of material, and if the claim of outperforming other protocols is fully supported by appropriate data. For example, Fig E6a shows comparable results between FASP and SP3 for duplicate aliquots of a yeast whole-cell lysate (10 μ g), but no data are provided for smaller amounts of material. It would be pertinent to make the comparison of identification between SP3 and FASP or StageTip (Kulak et al, 2014) for HeLa digests ranging from 500,000 to 1,000 cells.

Author response: Please see the response to Reviewer #1, point #5, as it addresses the same concern.

2. The description of the sample processing protocol is not entirely clear, and different parts of the workflow are scattered across multiple sections often applied to different samples. I was not able to find a clear step by step procedure on how to process sample with SP3 from cell lysis to MS analysis. A more straightforward description of the optimized SP3 protocol is required in the extended materials and methods section for users interested in applying this protocol for proteomics samples.

Author response: As outlined above, we have attempted to improve the detail and clarity of the SP3 methods throughout the main and expanded text. In addition, we have provided step-by-step details of the SP3 and related support protocols in the Expanded protocols.

3. Very often MS results are represented as overlaid XIC's. This comparison is visually attractive but not very informative. Can the authors provide label-free quantitation as scatter plots showing peptide intensities from different samples? These analyses can be performed using MaxQuant.

Author response: We appreciate the concerns of the reviewer in this aspect, and have gone back and forth on this point in previous iterations of the manuscript. In the end we feel that overlaid base-peak chromatograms provide the most detail in this instance. The data acquired related to the method optimization where the majority of the overlaid chromatograms are used were acquired on a low-resolution ion trap instrument in very short runs (30 or 60 minutes). Therefore, we would obtain very few peptide identifications from these data, limiting the value of a label-free intensity plot as suggested. Furthermore, we feel it is of value for the reader to be able to visualize properties such as hydrophobicity to assess potential bias between methods. These details are lost on label-free plots.

4. Can the authors provide data on sample binding capacity (protein and peptide) on magnetic beads? Different peptide/protein loadings should be provided to clearly illustrate what is the maximum binding capacity (preferable in the presence of different detergents to illustrate their effect). I'm surprised to see that the HILIC media has a binding capacity of 100 μ g of protein per 1 μ g of beads knowing that C18 media typically has a binding capacity of approximately 5% on a mass basis.

Author response: Data related to binding capacity is present in Figure E3a. The details of the experiment are discussed in the Expanded view methods section (Pg. 6 Para. 2). While we agree with the reviewer that capacity differences in the presence of different detergents is of interest, these details are beyond the scope of this manuscript and are better suited to a more focused audience and journal.